# Piezoresistive Multi-Walled Carbon Nanotube/Epoxy Strain Sensor with Pattern Design

**DOI:** 10.3390/ma12233962

**Published:** 2019-11-29

**Authors:** Mun-Young Hwang, Dae-Hyun Han, Lae-Hyong Kang

**Affiliations:** 1Department of Mechatronics Engineering, Jeonbuk National University, 567 Baekje-daero, Deokjin-gu, Jeonju-si 54896, Korea; 2LANL-JBNU Engineering Institute-Korea, Jeonbuk National University 567 Baekje-daero, Deokjin-gu, Jeonju-si 54896, Korea; 3Department of Flexible and Printable Electronics, Jeonbuk National University 567 Baekje-daero, Deokjin-gu, Jeonju-si 54896, Korea

**Keywords:** multi-walled carbon nanotubes, patterned type, polymer-matrix composites, strain sensing, piezoresistivity, structural health

## Abstract

Carbon nanotube/polymer-based composites have led to studies that enable the realization of low-cost, high-sensitivity piezoresistive strain sensors. This study investigated the characteristics of piezoresistive multi-walled carbon nanotube (MWCNT)/epoxy composite strain sensors subjected to tensile and compressive loads in one direction at relatively small amounts of strain. A patterned sensor was designed to overcome the disadvantage of the load direction sensitivity differences in the existing sensors. The dispersion state of the MWCNTs in the epoxy polymer matrix with the proposed dispersion process was verified by scanning electron microscopy. An MWCNT/epoxy patterned strain sensor and a patch-type strain sensor were directly attached to an acrylic cantilever beam on the opposite side of a commercial metallic strain gauge. The proposed patterned sensor had gauge factors of 2.52 in the tension direction and 2.47 in the compression direction. The measured gauge factor difference for the patterned sensor was less than that for the conventional patch-type sensor. Moreover, the free-vibration frequency response characteristics were compared with those of metal strain gauges to verify the proposed patch-type sensor. The designed drive circuit compensated for the disadvantages due to the high drive voltage, and it was confirmed that the proposed sensor had higher sensitivity than the metallic strain gauge. In addition, the hysteresis of the temperature characteristics of the proposed sensor is presented to show its temperature range. It was verified that the patterned sensor developed through various studies could be applied as a strain sensor for structural health monitoring.

## 1. Introduction

Although metallic strain gauges can be used to obtain stable measurements, properties like low flexibility and a non-embedded installation limit their application. Therefore, to replace metallic strain gauges in the structural health monitoring (SHM) field, much attention has been given to a sensor type capable of being mounted on a structure with a complex shape or embedded in a structure to improve SHM [1,2]. This has accelerated the research on smart materials [3,4,5,6]. Among the materials for new types of sensors, carbon nanotubes (CNTs) have been intensively studied for decades, and they have attracted great attention because of their superior mechanical and physical properties [7,8,9,10,11]. CNT-based composite sensors are also attracting much attention [12,13,14,15] because they are simple to design compared to structural sensors such as micro-electromechanical systems (MEMSs). Such structural sensors are expensive to manufacture and subject to limitations on the sensor geometry and materials used [16]. Composite sensors are relatively easy to fabricate and study because they provide direct measurements based on changes in electrical properties due to piezoelectricity and piezoresistivity. However, because there are various factors affecting the performance of the sensor, it is essential to accurately analyze these factors when fabricating the sensor. 

The critical factors in the design and fabrication of such composite sensors are the piezoresistivity and electrical conductivity, which affect the sensitivity of the sensor. The type of resin profoundly influences the piezoresistivity and viscosity of the resin used [17,18,19,20]. In addition, a CNT/polymer composite with an electrically percolated conductive filler has a high piezoresistivity at a relatively low CNT concentration [14,21,22]. Moreover, the conductivity of the CNT/polymer composite is temperature dependent. The change in the electrical resistance of a CNT/polymer composite with temperature is different from that for pure multi-walled carbon nanotubes (MWCNTs). In the CNT/polymer composite, decreases and increases in the electrical resistance with increasing temperature have both been reported. In composites with a high CNT filler concentration, the electrical resistance increased with increasing temperature [23,24]. The phenomenon can be explained by the thermal expansion mechanism of the resin and the tunneling effect between the CNTs. In composites with a low CNT filler concentration, the electrical resistance decreased with increasing temperature [25,26]. This phenomenon can be explained by the mechanism of thermal activation between the CNTs.

As previously mentioned, the characteristics simultaneously affect the sensitivity of the sensor. The intrinsic thermo-electrical properties and concentrations of CNTs, the electrical and mechanical properties of the polymer matrix, and the temperature range are significant parameters affecting the actual behavior of the composite. Therefore, the fundamental characteristics of a CNT/polymer composite sensor based on piezoresistivity should be studied considering the usage environment and design method for the sensor.

Although an MWCNT/epoxy composite has been studied, it has a disadvantage that makes it difficult to use as an SHM sensor. Typical MWCNT/polymer sensors have different sensor sensitivities in the tensile and compressive directions [13,14,15]. When a compressive force is applied to a sensor, the decrease in the distance between CNTs is limited because neighboring CNTs cannot penetrate each other. In contrast, when a tensile force is applied, the increase in the distance between adjacent CNTs is not limited. In other words, under a compressive strain, the piezoresistivity of the MWCNT/epoxy saturates with an increase in the applied strain. Thus, the sensitivity to a compressive strain is lower than that to a tensile strain. To develop a sensor that can be used in the SHM field based on the MWCNT/epoxy sensor already proposed by many researchers, a new sensor design is proposed and essential sensor characteristics are evaluated based on the shape of the developed sensor. Because a patterned sensor has been proposed to overcome the disadvantages of conventional MWCNT composite sensors, the changes in the essential characteristics of a patterned MWCNT/polymer sensor with a new shape are compared with those of a traditional sensor. First, a sensitivity comparison is performed in the tensile and compressive directions to evaluate the influence of the sensor structure. Second, an electrical resistivity evaluation in a high-temperature environment is carried out to investigate the temperature-dependent characteristics related to the filler and resin. Finally, the frequency response is investigated to confirm the availability compared to the conventional metal strain gauge in SHM.

## 2. Manufacture of Patterned Strain Sensor Using Piezoresistivity of MWCNT/Epoxy Composite

### 2.1. Design of MWCNT/Epoxy Patterned Sensor

The electrical conductivity of the MWCNT/epoxy is determined by the percolated concentration of networks of dispersed MWCNTs.

The MWCNTs do not need to be in contact with each other to transfer a charge within the insulating resin, but the maintenance of a nanometer distance (tunneling distance) between particles is required [13,15,27,28,29]. Therefore, in relation to the electrical properties of the MWCNT/epoxy, an appropriate dispersion state is necessary for the occurrence of the tunneling effect. The optimization process for the appropriate dispersion of the MWCNTs in the epoxy was studied using an experimental approach used in previous studies [30]. Although the distance between the MWCNTs with the best sensor performance at room temperature (20 °C) was obtained using the optimized process, the separation distance (Δd) between the MWCNTs according to temperature differed as a result of the thermal expansion or contraction of the polymer (shown in Appendix A).

The distance variation caused a change in the conductivity of the composite. Hence, the performance of the MWCNT/epoxy sensor changed. The operating mechanism of the composite strain sensor can be explained as a change in the contact resistance between the MWCNTs under tension or compression. There is no restriction on the distance when tension is applied. However, there is a restriction on the resistance change because the MWCNTs cannot penetrate each other when they are compressed (shown in Appendix A). For these reasons, typical MWCNT/polymer sensors have different sensor sensitivities in the tensile and compressive directions [28,30]. This study investigated a sensor shape, as shown in Figure 1, to compensate for the disadvantages of the conventional MWCNT/polymer sensors. The pattern format was applied to minimize the limitation of the CNT particle distance in the compression direction. Because of the structural advantages, when the transverse root is subjected to tension, the longitudinal root is compressed (Figure 1a). In the opposite case, the conditions mentioned are reversed (Figure 1b). In other words, with any external force, the ratio between the tension state and compression state is continuously maintained within the sensor, so that a constant resistance change and continuous performance are maintained. Further, because the material is made in a liquid form and cured, a simple process for applying or filling the sensor electrode to be formed can be used. This advantage can be retained because it is more economical and reliable than the method used for existing sensors.

### 2.2. Fabrication Process for MWCNT/Epoxy Patterned Strain Sensor

In this study, MWCNT/epoxy composites were fabricated using MWCNTs (CM-150, Hanwha Chemical, Seoul, Korea) and epoxy (KFR–120, KUKDO chemical, Seoul, Korea). Among the various polymer resins, epoxy resin was chosen because of its excellent adhesion to many substrates, as well as its high thermal and mechanical properties.

The purchased MWCNTs had a high aspect ratio of approximately 2 × 10^3^ and a high purity of 87–93 wt.%. The fabrication process for the patterned strain sensor developed in this study and the fabrication process for the patch-type strain sensor for comparison are shown in Figure 2. MWCNTs were dispersed in acetone for 1 h using an ultrasonic processor (Q125–220, Qsonica, Newtown, Connecticut, CT, USA). Then, the MWCNTs in the acetone were again dispersed using a high-speed mixer (8000D Mixer Mill, SPEX, Metuchen, New Jersey, NJ, USA) with a mixing speed of 1725 Revolutions Per Minute (RPM). The MWCNTs for the composite strain sensor were used without any further chemical treatment. The advantage of the solvent-based dispersion method without chemical treatment is that it maintains the inherent properties of the CNTs. MWCNTs with a concentration of 0.5 wt.% in the solvent were mixed with epoxy resin. The mixture was sonicated using the ultra-sonicator for 30 min. After that, a hardener (the specific mixing ratio of the epoxy and hardener was 100:27) was added to the MWCNT/epoxy. The final mixture was placed into a vacuum chamber to allow degassing. In this work, an acrylic beam was used as a cantilever test beam because of the high impact resistance needed for the impact experiment, its machinability, and its transparency, which made it easier to confirm the embedded forms. After the dispersion preparation, a constant pattern was made on the acrylic beam using a drilling machine, where a linear stage was used (shown in Appendix A) because the thickness of the pattern produced had to control the final thickness of the composite. The geometry of the pattern is shown in Figure 3. 

The MWCNT/epoxy mixture used to fill in the pattern was also cured for 8 h in a vacuum bag at 80 °C, as shown in Figure 4, to minimize the pores, which could interfere with complete curing and deteriorate physical properties. Because the mixture with more than 0.5 wt.% MWCNTs was very viscous, it was challenging to homogenously insert the mix into the formed pattern. Thus, the fabrication process using the vacuum bag assisted in keeping the inserted mixture homogenous because the vacuum bag ensured an even pressure in the profile. Before testing the sensing characteristics, the surface of each specimen was mechanically polished to minimize the influence of surface flaws, which mainly consisted of pores. In addition, to verify the advantages of the patterned type of sensor, as shown in Figure 5, patch-type sensors were made with 0.5 wt.% MWCNTs with thicknesses of 0.5, 1.5, and 2.0 mm using the same fabrication process, and their characteristics were also measured in the same environment. Conductive silver paste and conductive epoxy were attached to the ends of the sensors to measure the electrical characteristics, and the specimens were dried in a dry oven to increase the adhesion between the sensors and the silver paste. Various kinds of samples were made with three or more examples each to verify the reliability of the test results. Finally, commercial strain gauges were attached to the opposite side along the same axis as the MWCNT/epoxy nanocomposite on the beam to determine the gauge factor related to the applied strain on the cantilever beam. The change in the electrical resistance of the MWCNT/epoxy sensor as a function of temperature was measured with no load applied in a temperature range of 30–120 °C in the oven to investigate the temperature-dependent properties of the composite sensor. Each temperature step was maintained for at least 1 h.

## 3. Experimental Test

### 3.1. Experimental Setup for Determining Gauge Factor Using MWCNT/Epoxy Composite Sensor

A multimeter (8848-A, FLUKE, Everett, Washington, WA, US) was used to measure the variations in the resistance of the MWCNT/epoxy to determine the gauge factor of the MWCNT/epoxy composite strain sensor at room temperature and compare the strain-sensing ability to that of a commercial strain gauge. One end of the acrylic cantilever beam was fixed, and the free end of the beam was connected to the surface of the holder for the vertical load (shown in Appendix A). The strain measurement jig allowed the acrylic beam to deform up to 1 inch (deformation interval: 0.1 inch) in the vertical direction at the end to measure the low strain range (up to 2200 με) with tension force and compression force.

The gauge factor is given as the relation of the relative resistance change $\Delta R/R_{o}$ and applied strain:(1)$$K=\frac{\Delta R/R_{0}}{\varepsilon },$$
where K is the gauge factor, ε is the applied strain, ΔR is the change in resistance, and $R_{0}$ is the initial resistance, which is used to evaluate the relative resistance change of the strain gauge. To calculate the strain at the sensor measurement point from the deformation of the acrylic beam with a load at the end, Equation (2) was used, which was designed for a rectangular beam in the beam theory [31]:(2)$$\varepsilon =\frac{\sigma }{E}=\frac{3c\left(L-x\right)}{L^{3}}\delta ,$$
where *c* = t/2 is the distance from the neutral axis to the surface of the beam, *t* is the thickness of the beam, *L* is the length, *x* is the distance from the fixed end of the cantilever beam to the center of the sensor, and δ denotes the amount of deformation at the end with respect to the concentrated load at the end of the beam. Using Equations (1) and (2), the strain at the sensor measurement point was calculated with a constant amount of displacement generated from the fixed jig, and the gauge factor was calculated using the measured resistance change value.

### 3.2. Experimental Setup for Comparing MWCNT/Epoxy Sensor with Metal Strain Gauge

In this study, the frequency response of the proposed MWCNT/epoxy was compared with that of a commercial sensor. It was difficult to determine whether the developed sensor could sufficiently measure the strain in the structure using only the response to a static load. Therefore, if the frequency response of the MWCNT/epoxy were similar to those of the laser Doppler vibrometer (LDV) and commercial metal strain gauges, it would prove that the performance of the developed sensor was sufficient for use as a sensor. The free vibration signals generated by an impact hammer were analyzed by frequency response function measurement using data acquisition equipment (LMS SCADAS, SIEMENS, Munich, Germany) and its internal software (shown in Appendix A). Because the manufactured sensor had a high resistance, a high voltage was required for its operation. Thus, it was difficult to apply a general indicator to the MWCNT/epoxy sensor. Therefore, in this study, a Wheatstone bridge circuit designed for a drive voltage of 17 V (experimentally determined) was used (shown in Appendix A). A finely tunable variable resistor was used to create a dummy resistor for the developed sensor with high resistance at the Wheatstone bridge. An amplification gain of 10 dB was set to sufficiently measure a tiny resistance change in the MWCNT/epoxy composite. A 30 Hz filter circuit was added to solve the problem of noise amplification due to the high voltage and high amplification.

## 4. Results and Discussion

### 4.1. Morphology Analysis Using Scanning Electron Microscopy

The cut surfaces of MWCNT/epoxy patch-type sensors fabricated using the same process with various MWCNT concentrations were observed using scanning electron microscopy (SEM) to verify the dispersion ability of the proposed dispersion process. The specimens were sputter-coated with a thin layer of platinum to avoid charging during the analysis and to obtain a stable resolution. From the SEM images taken for the different MWCNT concentrations, as shown in Figure 6, it can be seen that the MWCNTs are randomly distributed in the epoxy polymer matrix, and the MWCNTs are dispersed homogeneously, other than some partial small agglomerations. These results demonstrated that the 0.5% solution prepared by the proposed process could be sufficiently dispersed in the pattern structure.

### 4.2. Sensing Property Analysis of Piezoresistive MWCNT/Epoxy Composite Sensor under Tensile and Compressive Loads

To determine the gauge factor and investigate the piezoresistive response of the MWCNT/epoxy composite strain sensor, the cantilever beam was deformed by tensile and compressive loads at room temperature. Each strain of the sensors at the measurement point in both the tensile and compressive directions was calculated using Equation (2). As shown in Table 1 and Figure 7, the relative resistance change $\Delta R/R_{o}$ was measured, and the gauge factors were determined from the slope between the strain (ε) and resistance change ($\Delta R/R_{o}$). 

In addition, the determined gauge factors were compared with the gauge factor (2.12) of the commercial metallic strain gauge, which was shown in the specification. Based on the results, the corresponding sensitivities of the MWCNT/epoxy patterned sensors under tensile and compressive loads were found to be 2.52 and 2.47, respectively. In other words, the patterned sensor exhibited a higher sensitivity than the conventional commercial strain gauge and patch-type sensor. As mentioned in the introduction, the results showed that for the patch-type sensor, the gauge factor in the compression load direction was smaller than that in the tension direction. Similar behaviors have also been observed in other studies [13,14,15]. However, in the case of the proposed patterned sensor, the difference in the gauge factors for the two directions was smaller than that for the patch type. As reported in many studies [32,33], CNT/polymer nanocomposites exhibit a nonlinear behavior at a high strain range with low CNT concentrations as a result of the dominant role of tunneling resistance, which tends to disappear with increasing CNT concentration. It has been demonstrated that the design of the patterned structure effectively controls the area of the sensor to maintain a constant tunneling effect. Further, based on the gauge factor of the patch-type sensor, the sensor has a higher sensitivity because the sensor is thinner. In other words, because the manufacturing method for the proposed pattern-type design can precisely fabricate a specific thickness, it is easy to manufacture a sensor with high sensitivity. Therefore, it was confirmed that the nonlinear behavior of the composite sensor could be controlled simply through the design of the sensor structure, even though it is manufactured with a relatively low MWCNT concentration. To be able to manage the sensitivity of the sensor on the micro-scale by controlling the production area of the sensor, a uniform dispersion is necessary, as shown in Section 4.1.

Figure 8 also shows the result of an experiment to measure the frequency response of the sensor by the free vibration caused by the impact hammer. Table 2 lists the results of a comparison of the primary frequency responses to verify the sensing ability of the patterned sensor.

Because the first mode frequency response of the MWCNT/epoxy strain sensor was the same as those of the commercial strain gauge and Laser Doppler Vibrometer (LDV) sensor, it is believed that the MWCNT/epoxy patterned strain sensor would be able to function sufficiently as a strain sensor. A numerical modal analysis was also conducted using the analysis program ANSYS (ANSYS Inc., Cannonsburg, PA, USA) to verify the measured data of the first frequency response and a similar first frequency response was obtained. It was also proven that the amplification circuit designed for high-resistance sensors was working correctly. The effect of the temperature on the electrical resistance of the MWCNT/epoxy composites was investigated by subjecting the specimens to three heating cycles in the temperature range of 30–80 °C, as shown in Figure 9 and Table 3. Cooling and heating were repeated in three cycles from 30–80 °C. The holding time of each temperature section was 2 h, and after reaching 80 °C, the specimen was naturally cooled until the sensor reached 30 °C.

Similar to other studies [33,34,35,36], where the resistivity increased upon heating due to the thermal expansion of the polymer matrix, here it was observed that the nanocomposite showed a repeatable positive temperature coefficient response under cyclic temperature changes when the temperature was less than 60 °C. However, when the temperature was increased from 60 °C to 80 °C, the resistance variation was saturated. This phenomenon was due to the limited thermal expansion of the resin as a result of the characteristics of the pattern structure, unlike the other patch-type sensors. When the MWCNT/epoxy sensor was cooled to 30 °C after being heated to 80 °C, the initial resistance of the sensor was restored. From these results, it was determined that the temperature range of the developed patterned sensor was up to 80 °C. The temperature coefficient resistance (TCR) of the developed sensor was 0.377 K^−1^ in the field up to 60 °C, where the resistance constantly increased. The high sensitivity of the piezoresistive nanocomposite to the temperature showed that MWCNTs/epoxy nanocomposites have multifunctional sensing capabilities because they are able to measure both strain and temperature.

## 5. Conclusions

In this study, a new type of strain sensor based on an MWCNT/epoxy composite was prepared using the patterned-type fabrication method. The MWCNT/epoxy patterned sensor with the same sensitivity to tension and compression had fewer limits on the form of the load and natural frequency of the structure compared to conventional sensors. Various experiments were conducted to investigate the piezoresistivity of the composite under tensile and compressive loads. SEM images confirmed the dispersion state of the MWCNTs in epoxy resin. The proposed dispersion process was also verified. A 0.5 wt.% MWCNT/epoxy composite was embedded in an embossed pattern in an acrylic cantilever beam. The resistivity changes were measured in the static state to investigate the piezoresistivity related to the gauge factor, which represented the sensing ability. A metallic strain gauge was also attached to the acrylic beam at the opposite side of the composite sensor to compare the sensing properties in a free vibration test. It was found that the patterned sensor showed relatively higher sensitivities than commercial strain gauges and patch-type sensors under tensile and compressive loads. It was also observed that the difference in the gauge factors in the patterned sensor was smaller than that in the patch-type sensor. These results verified the usability of the sensor. Based on the temperature measurements, it was found that the composite sensor showed a repeatable positive temperature coefficient behavior under a cyclic temperature range of 30–80 °C with a TCR of 0.377 K^−1^, which confirmed the multifunctional sensing ability of the nanocomposites. The MWCNT/epoxy sensor fabricated using the pattern method could be applied to real-time structural health monitoring. The MWCNT composite developed with the proposed fabrication method is advantageous because it can be miniaturized and mass-produced.

## Figures and Tables

**Figure 1 materials-12-03962-f001:**
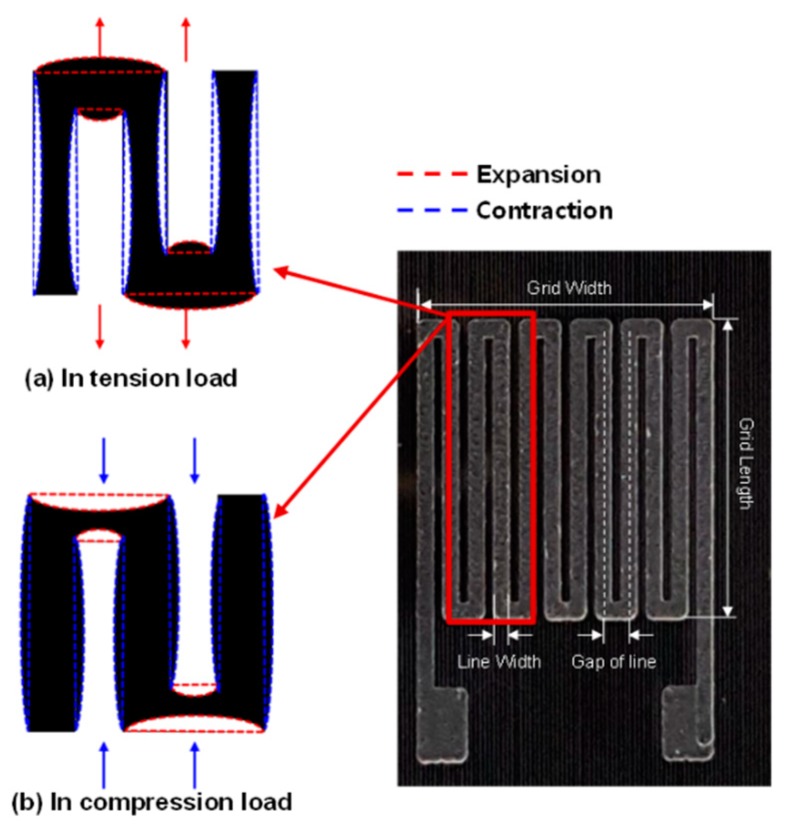
Patterned type of MWCNT/epoxy strain sensor that can minimize compression direction restriction with respect to CNT particle distance; (**a**) the pattern sensor under tension load, (**b**) the pattern sensor under compression load.

**Figure 2 materials-12-03962-f002:**
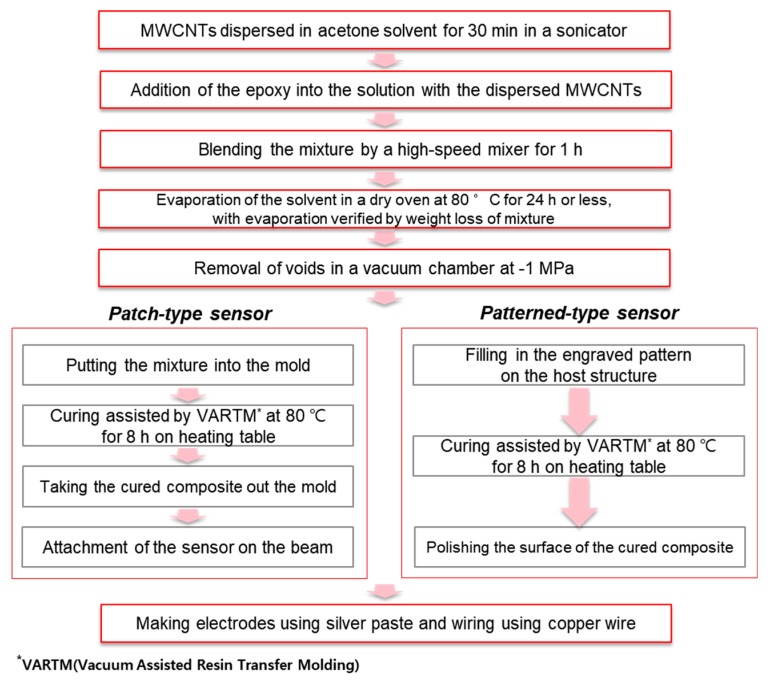
Designed fabrication process for the MWCNT/epoxy composite strain sensor.

**Figure 3 materials-12-03962-f003:**
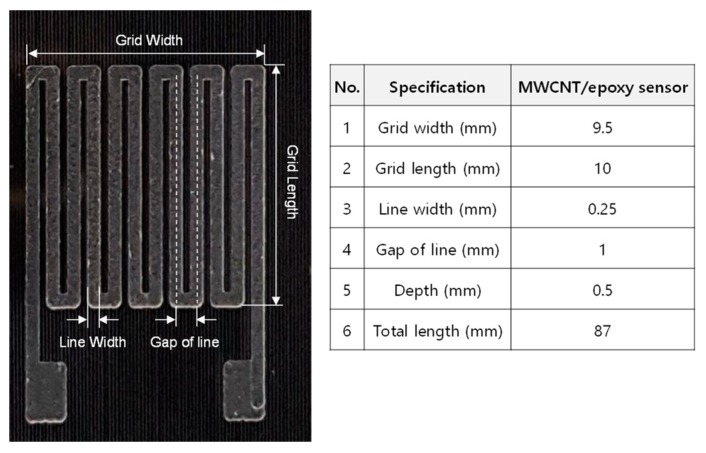
Geometry of patterned MWCNT/epoxy strain sensor.

**Figure 4 materials-12-03962-f004:**
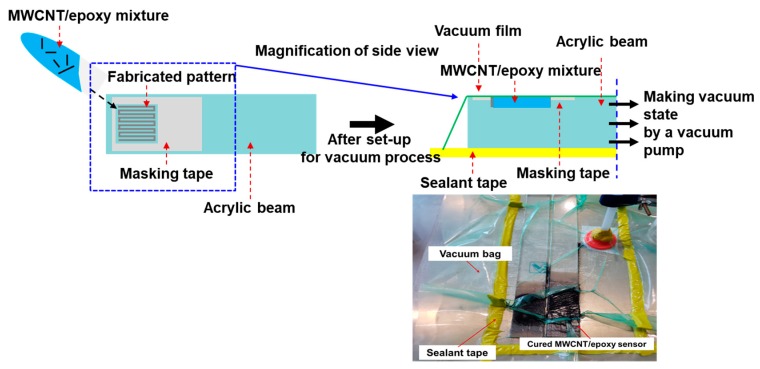
Curing process for MWCNT/epoxy strain sensor with pattern integrated into structure to remove pores in sensor.

**Figure 5 materials-12-03962-f005:**
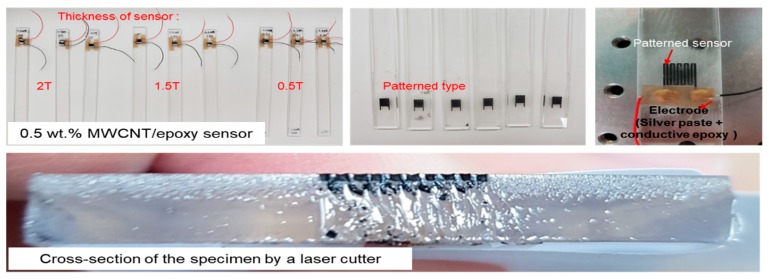
Patterned and patch-type specimens of MWCNT/epoxy strain sensors used to measure sensing characteristics.

**Figure 6 materials-12-03962-f006:**
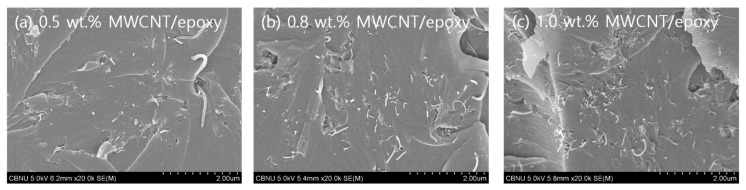
Patch-type sensors manufactured with various concentrations of MWCNTs to verify the proposed dispersion process: (**a**) 0.5 wt.%, (**b**) 0.8 wt.%, and (**c**) 1.0 wt.%.

**Figure 7 materials-12-03962-f007:**
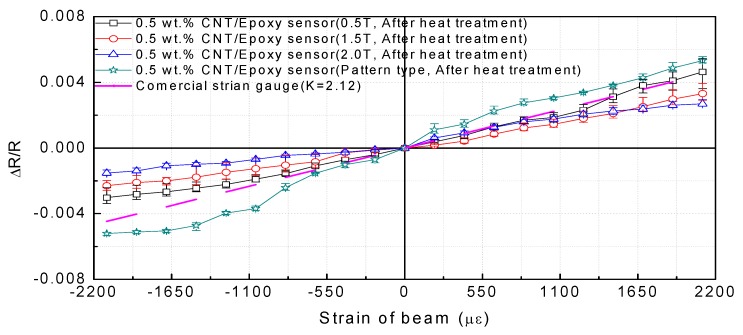
Results for resistance changes according to tip deflection of cantilever beam.

**Figure 8 materials-12-03962-f008:**
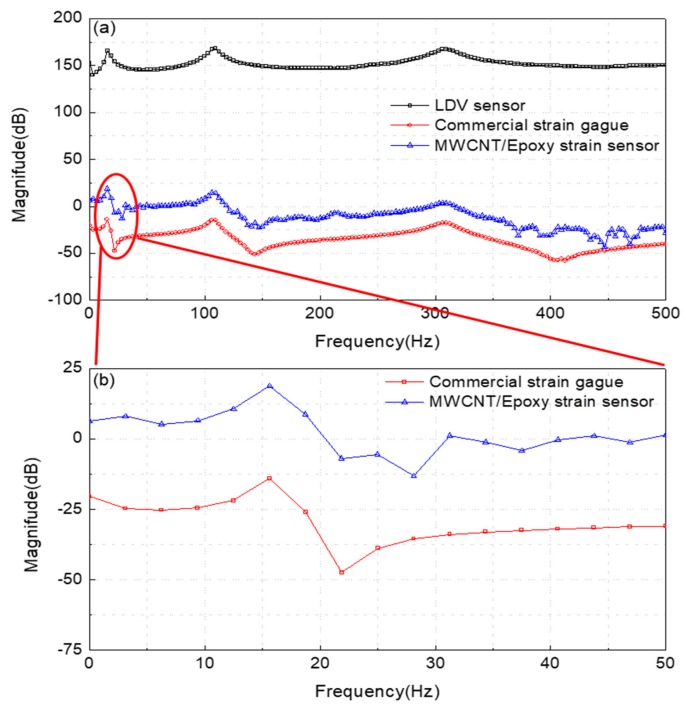
Results of experiment to measure frequency response of sensor by impact: (**a**) results in 0–500 Hz range and (**b**) results in 0–50 Hz range.

**Figure 9 materials-12-03962-f009:**
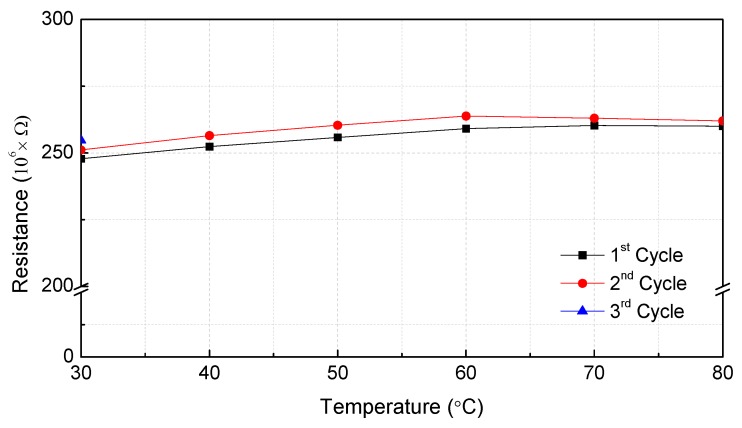
Relative resistance deviation of MWCNT/epoxy patterned composite strain sensor under cyclic temperature change effect at 30–80 °C for three cycles.

**Table 1 materials-12-03962-t001:** Gauge factors of 0.5 wt.% MWCNT/epoxy strain sensors according to fabrication type and load direction.

Thickness (mm)	Gauge Factor in Compression Direction	Gauge Factor in Tension Direction
0.5	1.43	2.20
1.5	1.08	1.57
2.0	0.72	1.14
Pattern (0.5)	2.47	2.52

**Table 2 materials-12-03962-t002:** Results of comparison of primary frequency responses to verify sensing ability of patterned sensor.

Analysis (ANSYS)	MWCNT/Epoxy Patterned Strain Sensor	Commercial Strain Gauge	LDV Sensor
15.11 Hz	15.63 Hz	15.63 Hz	15.63 Hz

**Table 3 materials-12-03962-t003:** Relative resistance deviation of MWCNT/epoxy patterned composite strain sensor under cyclic temperature change.

Temperature (°C)	Resistance (×MΩ)	Measurement Interval (h)	Temperature (°C)	Resistance (×MΩ)	Measurement Interval (h)
30	247.8	0	30	251.1	24
40	252.4	2	40	256.5	2
50	255.8	2	50	260.3	2
60	259.1	2	60	263.8	2
70	260.3	2	70	263.0	2
80	260.0	2	80	262.1	2
			30	254.8	24

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
