# Peer review of "Piezoresistive Multi-Walled Carbon Nanotube/Epoxy Strain Sensor with Pattern Design"

_materials, 2019, doi:10.3390/ma12233962_

Round 1
Reviewer 1 Report
The paper presents a new strain sensor based on a patterned epoxy doped with multi walled carbon nanotubes. Experiments are presented to demonstrate the effectiveness of the sensor and its thermal drift. Overall, the paper is interesting but requires a significant English check. Furthermore the paper should more clearly address:
-the novelty of the sensor with respect to existing solutions
-the advantage of using nanotubes (gauge factor is low)
The SEM inspection seem to highlight a very small number of nanotubes in a representative volume element. Is the fraction of nanotubes enough to boost piezoresistivity? Did the authors perform any percolation study?
Does the considered material suffer from polarization issues? The authors may consider the following study where a measurement technique for composites with MWCNTs was developed to get rid of polarization:
Downey, A., D'Alessandro, A., Ubertini, F., Laflamme, S., Geiger, R., Biphasic DC measurement approach for enhanced measurement stability and multi-channel sampling of self-sensing multi-functional structural materials doped with carbon-based additives (2017), Smart Materials and Structures, Volume 26, Issue 6, Article number 065008.
The use of MWCNT can provide dieletric materials with piezoresistive and crack sensitive properties. The authors may consider the following paper and reference therein to enlarge their literature review:
Downey, A., D'Alessandro, A., Baquera, M., García-Macías, E., Rolfes, D., Ubertini, F., Laflamme, S., Castro-Triguero, R., 2017, Damage detection, localization and quantification in conductive smart concrete structures using a resistor mesh model, Engineering Structures, Volume 148, Pages 924-935
Reviewer 2 Report
Review Report – Materials
Manuscript ID: materials-643883
Title: Study on Sensing Characteristics of Multi-Walled Carbon Nanotube Epoxy Patterned Type Strain Sensor Based on Piezoresistive Effect
Authors: Mun-Young Hwang, Dae-Hyun Han and Lae-Hyong Kang*
In this work, the Authors have presented the results of new MWNCT/epoxy composite patterned and patch strain sensors subjected to tensile and compressive loads at relatively small amounts of strain. The dispersion state of MWCNTs in epoxy resin is confirmed. Patterned type sensor showed higher sensitivities than commercial strain gauges as well as the patch type sensors under tensile and compressive load. Repeatable positive temperature coefficient behavior under cyclic temperature ranging (30-80 °C) confirms the multifunctional sensing ability of nanocomposites.It is suggested that the new MWCNT/epoxy sensor made by patterned method is potential candidate to the real-time structural health monitoring.
General remarks
The idea seems very interesting an promissing. In my opinion, the quality of the manuscript and presented results are appropriate for to be published in this Journal. However, in the light of scientific content, there are some points which I feel should be addressed and clarified:
Could the Aurhors please comment on the reproducibility of sensitivity factor in their study?the variability of resistance in the bach of samples. Please see references Michelis, F., Bodelot, L., Bonnassieux, Y., & Lebental, B. (2015). Highly reproducible, hysteresis-free, flexible strain sensors by inkjet printing of carbon nanotubes. Carbon, 95, 1020-1026. Bodelot, L., Pavić, L., Hallais, S., Charliac, J., & Lebental, B. (2019). Aggregate-driven reconfigurations of carbon nanotubes in thin networks under strain: in-situ characterization. Scientific reports, 9(1), 5513. I also suggest editing of English language and style.
Reviewer 3 Report
The authors report a study of Multi-Walled Carbon Nanotube Epoxy Strain Sensor based on Piezoresistive Effect. While the study is interesting and the results are significant, here are my comments:
The title of the manuscript is too long. It can be modified to read as follows:
"Piezoresistive Multi-Walled Carbon Nanotube Epoxy Strain Sensor".
Shorter title can convey the same meaning as the authors would like it.
The language can be improved throughout the manuscript. The number of figures can be reduced by including only those figures that are critical to the results of the study For example, Figures 1,2,5,9, 10 and 11 can be included in supplementary materials and are not necessary to be published in the Journal.
In Table 3, "Measurement" should appear in one line.
What are the error bars in the measurements?
How many samples were made? Were the measurements repeatable? reproducible? It would help to add a schematic of the cross section of the device.
Round 2
Reviewer 1 Report
The authors have revised the paper in a proper way and provided appropriate responses to this reviewer's comments. Publication is now recommended.